# A Comparison of Established Diagnostic Criteria for Cachexia and Their Impacts on Prognostication in Patients with Oesophagogastric Cancer

**DOI:** 10.3390/cancers17030448

**Published:** 2025-01-28

**Authors:** Leo R. Brown, Maria Soupashi, Michael S. Yule, Cathleen M. Grossart, Donald C. McMillan, Barry J. A. Laird, Stephen J. Wigmore, Richard J. E. Skipworth

**Affiliations:** 1Clinical Surgery, University of Edinburgh, Royal Infirmary of Edinburgh, Edinburgh EH16 4SA, UKrichard.skipworth@nhs.scot (R.J.E.S.); 2Institute of Genetics and Cancer, Western General Hospital, University of Edinburgh, Crewe Road South, Edinburgh EH4 2XU, UK; 3St Columba’s Hospice, Edinburgh EH5 3RW, UK; 4Academic Unit of Surgery, University of Glasgow, Royal Infirmary, Glasgow G4 0SF, UK

**Keywords:** cachexia, oesophageal cancer, gastric cancer, malnutrition

## Abstract

Cachexia is a complex syndrome, characterised by the loss of muscle and fat, which commonly affects patients with cancer. It is particular common in patients with tumours of the oesophagus and stomach. This paper compares two definitions for cachexia and their associations with survival in patients with oesophagogastric cancers. Across 465 patients identified during a 2-year study period, it was noted that differing definitions led to different patients being diagnosed as cachectic. Less than half of those patients who met one definition were also cachectic when the other was used. The presence of cachexia diagnosed using the more contemporary GLIM criteria was more strongly associated with poor survival. This definition requires patients to demonstrate an inflammatory response, which is known to be a key feature of cachexia. The GLIM criteria should be used in clinical and research settings to identify patients with cancer cachexia. This will allow improved risk stratification and targeted intervention for this high-risk patient group.

## 1. Introduction

Despite advances over recent decades, oesophageal and gastric cancer remain associated with a notably poor prognosis. Together these diseases account for approximately 5% of all malignancies diagnosed annually in United Kingdom, with the majority presenting at an advanced stage [1,2]. Many patients are found to have metastatic spread at the time of first diagnosis, and the affected population are often elderly and comorbid. As such, few patients are ever amenable to ’curative’ therapy and, amongst those who are, this is associated with a considerable risk of morbidity [3,4] and mortality [5].

Cancer cachexia is highly prevalent in patients with oesophagogastric (OG) cancer [6], including in those with curatively treated disease [7]. This multifactorial syndrome, characterised by tissue wasting, anorexia, and fatigue, is far more complex than inadequate nutritional intake and is the result of complex interactions between the tumour and its host. Systemic inflammation, alterations to metabolism, and endocrine dysfunction all drive the involuntary loss of skeletal muscle and fat [8]. This is not reversible with nutritional support alone and leads to progressive functional decline. Cachexia is associated with greater risks of chemotherapy-related toxicities and dose reduction [9], poorer quality of life [10], and shorter survival [11].

DeWys et al. first studied the prognostic effect of weight loss amongst patients undergoing chemotherapy in the 1970s and noted its strong association with reduced survival [12]. This work set a precedent for weight loss being considered the key clinical feature in early diagnostic criteria for cachexia. In 2011, a consensus definition for cancer cachexia was published by Fearon et al. [13]. This landmark paper has been instrumental for guiding subsequent cachexia research over the last decade. More recently, experts from clinical nutrition and medical communities convened as part of the Global Leadership Initiative on Malnutrition (GLIM) to provide an updated consensus on the diagnosis of malnutrition [14]. As part of this, the group suggested an alternative definition of cachexia, describing it as disease-related malnutrition in the presence of systemic inflammation [15]. It has become increasingly evident over time that inflammation is a key driver of cachexia’s pathophysiology, and the new GLIM definition requires patients to not only demonstrate phenotypical changes but also an aetiological driver. This allows distinction to be made between cachexia and ‘starvation type’ malnutrition. The GLIM definition has now been utilised as part of guidance from the European Society of Medical Oncology (ESMO) [16] and the European Society for Clinical Nutrition and Metabolism (ESPEN) [17]. Both Fearon and GLIM are consensus-derived definitions and there remains a paucity of real-world data examining their prognostic influence in cancer populations. While previous studies have noted an adverse effect of GLIM-defined malnutrition in oesophagogastric cancer [18], its proposed application in defining cachexia remains underexplored, and no comparisons to the consensus definition have yet been completed.

The adverse impact of cachexia in oesophageal cancer is irrefutable; however, the lack of consensus regarding how best to define it poses challenges for the standardisation of research and implementation into clinical practice. Therefore, the primary aim of this study was to compare the prognostic utility of the two most established contemporary cachexia definitions (Fearon et al. and GLIM) in patients with oesophagogastric cancers of all stages. The secondary aim was to compare the influences of the three potential phenotypical criteria of the GLIM definition on survival.

## 2. Methods

This multi-centre observational cohort study retrospectively utilised patient-level data collected during routine clinical care. Ethical review and approval were waived for this study due to this being a retrospective analysis of routinely collected clinical data. Local institutional approval (Caldicott Guardian) was obtained prior to data collection. The manuscript is reported in line with STROBE guidance [19].

### 2.1. Patient Cohort

Consecutive patients (≥18 years) were identified from the prospectively maintained database of the South-East Scotland Cancer Network. This is one of the three regional cancer networks in Scotland, serving a population of approximately 1.5 million across four NHS Scotland Health Boards. All patients were assessed at the specialist upper gastrointestinal cancer multidisciplinary team (MDT) with newly diagnosed cancer of the oesophagus, gastro-oesophageal junction (GOJ), or stomach between 1 January 2019 and 31 December 2020. Patients were included irrespective of treatment pathway or modalities. Any patients with a non-malignant diagnosis, a concurrent malignancy of an alternative type, or recurrence of a previously diagnosed OG cancer were excluded.

### 2.2. Clinical Staging Investigations

All patients underwent a diagnostic upper gastrointestinal endoscopy and contrast-enhanced thoraco-abdominal staging computed tomography (CT) scan unless investigation was contraindicated by either frailty or comorbidity. Disease stage was reported as per TNM classification (8th edition) [20]. The Siewert classification was used to categorise GOJ tumours based on endoscopic appearance [21]. Tumour site was dichotomised with type I/II GOJ tumours considered to be oesophageal and type III tumours considered to be gastric. Additional staging investigations, such as positron emission tomography (PET) scans or diagnostic laparoscopy with cytology of peritoneal washings, were performed selectively based on local guidance.

### 2.3. Clinicopathological Data Collection

Clinicopathological data were collated from the South-East Scotland Cancer Network database with additional datapoints sought from electronic patient records. These included basic demographics, American Society of Anaesthesiologists (ASA) grade [22], Charlson comorbidity index [23], Eastern Cooperative Oncology Group Performance Status (ECOG-PS) [24], height and weight measurements, and disease characteristics. Pre-diagnosis recorded weight loss, or patient-reported involuntary weight loss, was also retrieved. Haematological blood results, obtained at clinical staging (prior to commencement of any treatment), were obtained for all included patients. Neutrophil lymphocyte ratio (NLR) was calculated by dividing the absolute neutrophil count by the absolute lymphocyte count. Details regarding treatment intent and modalities were also retrieved alongside postoperative outcomes and dates of death or last known follow-up.

### 2.4. Body Composition Analysis

Portal venous-phase CT scans of the chest and abdomen, obtained at clinical staging (prior to commencement of any treatment), were retrieved for all included patients. Data Analysis Facilitation Suite version 3.7 (DAFS ^®^ Voronoi Health Analytics Inc., Vancouver, BC, Canada, 2021) was used for body composition analyses. Cross-sectional area (cm^2^) of skeletal muscle, at the mid-3rd lumbar vertebral (L3) level, was normalised for height (m^2^) to calculate skeletal muscle index (SMI) (cm^2^/m^2^). Automated vertebral level annotation and tissue segmentation were cross-checked by a trained clinician investigator with manual edits completed as necessary. Imaging quality was also assessed at this point and data were not retrieved for any scans where artefact or other issues precluded reliable body composition analysis.

### 2.5. Cachexia Definitions

Patients in this study were compared based on whether or not they met the below criteria for cancer cachexia at the time of diagnosis.

Fearon et al. [13] diagnosed cachexia in patients who, in the preceding 6 months, had experienced weight loss >5% OR weight loss >2% with BMI <20 or low muscle mass.

The GLIM criteria [14] required ≥1 phenotypical and ≥1 aetiological criteria for a diagnosis of malnutrition. Phenotypical criteria were weight loss >5% (<6 months) or >10% (>6 months), BMI <20 (<70 years) or <22 (>70 years), OR reduced muscle mass. Aetiological criteria for malnutrition were either reduced food intake/assimilation OR inflammation. For a diagnosis of cachexia, patients had to meet the GLIM consensus criteria for disease-related malnutrition with inflammation [15]. As such, patients who met ≥1 phenotypical criteria with systemic inflammation were classified as cachectic.

For each definition of cachexia, CT body composition derived SMI was used to assess muscle mass. Thresholds for low SMI were defined as per Martin et al. (<53 cm^2^/m^2^ [BMI ≥ 25.0] or <43 cm^2^/m^2^ [BMI < 25.0] for males and <41 cm^2^/m^2^ for females) [25]. NLR was selected as a suitable marker of systemic inflammation that was collected as part of routine clinical practice for this patient group. While cut-offs for ‘high’ NLR vary in the existing literature [26], a validated threshold of ≥3.5 was defined a priori for this study [27]. If missing data precluded classification based on either set of criteria, patients were excluded (Appendix A).

### 2.6. Outcome Measures

The primary outcome of interest in this study was overall survival. All included patients had a minimum of 3-years follow-up from the date of their diagnosis. Secondary outcomes, for those treated via surgical resection, included postoperative length of stay; complication rate, graded as per Clavien-Dindo [28]; and pathological outcomes.

### 2.7. Statistical Analyses

Data analyses were undertaken using R 4.3.0 (R Foundation for Statistical Computing, Vienna, Austria) with *tidyverse*, *finalfit*, *eulerr, mice*, *survival*, and *survminer* packages. Categorical data were cross-tabulated, and differences were assessed with χ^2^ or Fisher’s exact test. Continuous data were presented as either means with standard deviation (SD) or medians with inter-quartile range [IQR], based on visual and statistical evaluation for normality. Subsequent testing was performed via appropriate parametric or non-parametric tests. Multivariable regression, via Cox proportional hazard modelling, estimated the association between each cachexia classifier and overall survival. Plausible confounders were explored with preliminary modelling and first-order interactions were checked. Variables included in the final regression model were selected based on minimisation of the Akaike information criterion (AIC) and maximisation of the c-statistic. Multiple imputation by chained equations (MICE) was undertaken for missing data with analyses performed across 10 multiply imputed datasets and pooled using Rubin’s rules [29]. 

## 3. Results

A total of 611 patients were retrieved from the source dataset. Patients were excluded if presenting with the recurrence of a previously diagnosed OG cancer (*n* = 40) or if they had a pre-existing or concurrently diagnosed malignancy of a different type (*n* = 12). Included patients were required to have all of the necessary datapoints for assessment across the considered diagnostic criteria. As such, exclusions were required for patients without the documented quantification of any involuntary weight loss (*n* = 53) or those who did not have a recorded full blood count (*n* = 7). The absence (*n* = 12) or insufficient quality (*n* = 22) of the staging thoraco-abdominal CT scans that precluded body composition analysis was a further reason for ineligibility. The resultant final cohort included 465 patients (Appendix A).

Overall, 311 males (66.9%) and 154 females (33.1%) who were diagnosed with oesophageal (72.0%) or gastric cancer (28.0%) across the 2-year study period were evaluated. The median age at diagnosis was 71 [IQR: 63–79] years with a range from 34 to 95 years. Across the cohort, 59.1% (*n* = 275) met the Fearon et al. diagnostic criteria for cancer cachexia and 44.1% (*n* = 205) were cachectic as per the GLIM criteria at diagnosis.

One hundred and forty-three patients met neither of the two considered definitions for cachexia (30.8%). While overlap was evident amongst the other 322 patients who did meet at least one of the diagnostic criteria, only 49.1% of these met both (Figure 1). All patients who were cachectic according to Fearon et al., but did not meet the GLIM criteria (*n* = 117), failed to do so owing to the biochemical absence of an inflammatory response. There were 47 patients who met the GLIM criteria for cachexia but were not cachectic according to Fearon et al.

### 3.1. Clinicopathological Characteristics

The clinicopathological characteristics at diagnosis were compared between cachexia definitions in Table 1. A higher median age was seen amongst patients who met the Fearon (73 vs. 69 years) and GLIM (74 vs. 68 years) diagnostic criteria for cachexia (both *p* < 0.001). Sex distributions were comparable throughout. While no differences in comorbidity were noted between groups with the Fearon definition, patients who were cachectic according to the GLIM criteria had a significantly higher ASA grade (*p* = 0.005) and Charlson comorbidity index (*p* = 0.012). A greater proportion of patients who met the GLIM diagnostic criteria for cachexia had a poorer ECOG-PS (*p* < 0.001). Significantly lower staging weight, BMI, and greater pre-diagnosis involuntary weight loss values were evident in patients who met either cachexia classifier (both *p* < 0.001).

A greater proportion of patients who met the Fearon diagnostic criteria (*p* = 0.008) had gastric, rather than oesophageal, tumours. Both definitions were associated with an advanced cT stage (Fearon: *p* = 0.022 and GLIM: *p* < 0.001) and a greater proportion of patients with GLIM-defined cachexia had advanced nodal disease (*p* = 0.001) when compared to those with Fearon-defined cachexia. Distant metastatic disease was also associated with GLIM criteria (Fearon: 26.9% vs. GLIM: 49.8%, *p* < 0.001).

Associations between cachexia definitions and clinical characteristics were less evident upon the subgroup analysis of curative (Appendix B) and non-curatively treated subgroups (Appendix C). Patients with incurable disease who met the GLIM criteria for cachexia were older (*p* = 0.017) and had a poorer ECOG-PS (*p* = 0.013). Similar associations were not evident amongst patients with Fearon-defined cachexia. Clinical tumours and nodal stages were comparable across both classifiers amongst the curatively treated subgroup. GLIM cachexia was associated with an advanced cT stage amongst those with incurable disease (both *p* = 0.020).

### 3.2. Overall Survival

The median survival across the overall cohort was 257 days (95% CI: 226–306). This was significantly reduced in the group of patients who met either Fearon or GLIM definitions for cachexia (median: 179 days [95% CI: 152–226]) when compared to those who met neither (median: 503 days [95% CI: 428–795], *p* < 0.001) (Figure 2A*)*.

Survival was decreased in patients who met the Fearon et al. diagnostic criteria for cachexia (median: 183 days [95% CI: 150–234]) relative to those who did not (median: 420 days [95% CI: 329–505], *p* < 0.001) (Figure 2B). A significant association with poorer survival was also observed for patients who met the GLIM criteria for cachexia (median: 135 days [95% CI: 108–171] vs. 446 days [95% CI: 383–515], *p* < 0.001) (Figure 2C*)*. Survival in patients who met the Fearon et al. diagnostic criteria only (median 363 days) was significantly lower than that of those met neither definition of cachexia (median: 503 days, *p* = 0.003) (Table 2 & Figure 2D). The median survival amongst those who met the GLIM criteria in isolation (168 days) was notably lower and comparable to that of patients who met both diagnostic criteria (median: 120 days, *p* = 0.300).

On comparison between tumour sites, decreased survival was evident for both classifiers amongst both the subgroups of patients with oesophageal cancer (both *p* < 0.001) and those with gastric cancer (Fearon cachexia: *p* = 0.028 and GLIM cachexia: *p* < 0.001, Appendix D).

Amongst the non-curative subgroup (*n* = 331), both definitions of cachexia were associated with poorer overall survival (Fearon cachexia: *p* = 0.001 and GLIM cachexia: *p* < 0.001, Appendix E). However, amongst those patients who were treated with curative intent (*n* = 134), no significant association with overall survival was evident for either definition (Fearon cachexia: *p* = 0.270, or GLIM cachexia: *p* = 0.800), although the cachectic proportions within this subgroup were small.

### 3.3. Effect of Cachexia Classifiers on Overall Survival

Two separate multivariable cox regression models were constructed to analyse the association between each cachexia definition and overall survival following adjustment for relevant confounding variables (Table 3 and Table 4). A poorer ECOG-PS and nodal and metastatic disease were clinicopathological characteristics consistently associated with reduced overall survival across both models. Fearon et al.-defined cachexia (aHR: 1.41 [95% CI: 1.13–1.76], *p* = 0.002) was associated with shorter overall survival. However, cachexia defined using the GLIM criteria was found to have a greater adverse prognostic effect (aHR: 1.57 [95% CI: 1.25–1.96], *p* < 0.001).

### 3.4. Postoperative Outcomes

Of the 134 patients treated with curative intent, 87 had surgery following neoadjuvant chemo(radio)therapy and a further 16 went directly to operative management (Appendix A). One hundred and three patients underwent surgical resection via oesophagectomy (Two-Stage/Ivor Lewis (*n* = 71) or Three-Stage/McKeown (*n* = 3)) or gastrectomy (total (*n* = 17) or subtotal (*n* = 12)). Five patients had an open/close laparotomy owing to unresectable disease that had not been identified on preoperative imaging.

Amongst the operative subgroup, 42.7% met the Fearon et al. criteria (*n* = 44) and 17.5% (*n* = 18) met the GLIM criteria for cachexia (Appendix F). Postoperative lengths of stay, complication rates, proportions of R1 resections, and pathological tumour (pT) stages were all comparable between groups using each of the definitions.

### 3.5. Comparison of GLIM Phenotypical Criteria

Patients who met the GLIM criteria (*n* = 205) underwent a secondary analysis to assess whether differences were evident based on which of the three phenotypical criteria were met. Of these, 20.0% (*n* = 41) met all three criteria and 40.0% met two out of three. Low muscle mass was the only phenotypical criterion found in 28.3% of patients and weight loss was the only criterion for 11.7%. No patients met the GLIM diagnostic criteria by low BMI in the absence of other phenotypical characteristics for malnutrition.

Survival was comparable between patients who met the weight loss criteria and those who did not (*p* = 0.190, Appendix G). However, both low BMI values (*p* = 0.001) and low radiological muscularity (*p* = 0.015) were associated with significantly shorter overall survival. Following adjustment for confounding variables, neither weight loss (aHR: 1.00 [95% CI: 0.72–1.40], *p* = 0.977) nor low BMI (aHR: 1.05 [95% CI: 0.71–1.57], *p* = 0.791) was associated with reduced survival (Appendix H and Appendix I). However, an adverse effect was evident for patients who met the GLIM criteria for cachexia with low radiological muscularity (aHR: 1.88 [95% CI: 1.15–3.07], *p* = 0.012) (Appendix J).

## 4. Discussion

The findings of the present study confirmed that cachexia is highly prevalent in patients newly diagnosed with oesophagogastric cancer. This was evident in both curative and non-curatively treated patient groups; however, rates were higher in those with incurable disease. Prevalence was comparable with previous estimates [7] but varied markedly based on the definition used, with the GLIM defining fewer patients (44.1%) than Fearon et al.’s criteria (59.1%). While overlap was evident, less than half of those who were cachectic using one of the considered definitions also met the other diagnostic criteria for the other. As such, it is highly likely that papers that use different cachexia classifiers are not describing comparable patient cohorts.

Cachexia based on the GLIM criteria was diagnosed in 44.1% of patients. While many of these patients (*n* = 158) also met the Fearon diagnostic criteria, the absence of a systemic inflammatory response precluded a GLIM diagnosis in the remainder of the Fearon et al. subgroup (*n* = 47). Patients with GLIM-defined cachexia demonstrated the greater adjusted magnitude of association with poor survival (aHR: 1.57 (95% CI: 1.25–1.96)) of the two considered definitions. Furthermore, on the sub-analysis of GLIM-cachectic patients, low muscle mass appeared to carry a greater prognostic effect than a low BMI or weight loss. Of note, patients who met the Fearon et al. definition, but not the GLIM criteria, had an overall survival approximately 40% longer than the median of the overall cohort. In contrast, survival in those patients who only met the GLIM diagnosis of cachexia was approximately 40% shorter. This is highly suggestive that the negative prognostic effect of meeting the Fearon diagnostic criteria for cachexia is driven by the subset of patients who also meet the GLIM criteria.

The 2011 Fearon et al. consensus paper is the most widely cited within the literature to date. Indeed, recent pioneering cachexia studies, such as the phase II trial of ponsegromab [30], have used Fearon’s definition in their inclusion criteria. However, following the 2019 publication of the GLIM, several influential organisations such as ESMO and ESPEN have utilised the more contemporary definition in their guidance [17,18]. Such choices between the two definitions are being made in the absence of real-world evidence comparing them and their relationship with outcomes of interest. Evidence regarding how best to define cachexia is urgently required to ensure the homogeneous and effective characterisation of the most clinically relevant patients.

Vanhoutte et al.’s 2016 paper is one other within the current literature that has compared definitions of cachexia and their association with survival [11]. The group undertook a series of assessments across a mixed cohort of 167 patients with gastrointestinal, lung, breast, or head/neck cancers. Of these, ~70% met Fearon et al. (2011) diagnostic criteria and ~40% met the Evans et al. (2008) criteria during the study period. Their results showed that both Evans et al. (*p* < 0.001) and Fearon et al. (*p* = 0.034) definitions were associated with shorter overall survival, but a greater difference was evident with Evans et al.’s (HR 3.32 (95% CI: 2.15–5.14)) than with Fearon et al.’s criteria (HR 1.82 (95% CI: 1.19–2.77)). The key differences between the two are Evans et al.’s considerations of symptoms (fatigue and anorexia), function (decreased muscle strength), and biochemistry (CRP/albumin/haemoglobin) in addition to weight loss. These extra criteria are certainly relevant to the cachectic phenotype, and it is unsurprising that they may provide additional prognostic value. However, the symptoms included in these criteria are quite subjective and other elements may be not assessed as part of routine clinical care (e.g., hand grip strength for muscle function). As such, the reliable application of the Evans et al. definition of cachexia is particularly challenging across retrospective cohorts. This is likely to be a key reason for it featuring less frequently within the literature, and indeed for its exclusion from this study. However, in the research setting, we must aspire towards a more robust, holistic, and prospective assessment of cachexia, and ongoing studies such as the REVOLUTION (NCT04406662) and REVOLUTION Surgery (NCT05642819) studies will help develop our understanding of its complex phenotype.

In upper gastrointestinal tract cancers, the effects of tumour burden (dysphagia/obstruction/early satiety) on oral intake can put patients at particular risk of ‘starvation-type’ malnutrition. As such, definitions that rely on the BMI and/or degree of weight loss in isolation (e.g., Fearon et al.) may be less able to accurately distinguish between patients experiencing wasting-cachexia-driven catabolic changes and those starved patients. While the presence of malnutrition of any aetiology is likely to negatively affect outcomes in patients with cancer, ‘starvation-type’ malnutrition should be reversible with adequate nutritional support while cachectic wasting will not [13]. Dietetics is a key component of the multidisciplinary treatment approach offered to patients in our region with upper gastrointestinal cancer. The service routinely screening for malnutrition in this patient group and nutritional support is likely to have been initiated at an early stage. This may have ameliorated the adverse effect of starvation but would be less influential in treating those with ‘true’ cachexia. If data pertaining to nutritional intake and assimilation were available, it would be informative to compare the outcomes of patients with GLIM-defined cachexia to those of patients with GLIM-defined malnutrition (without inflammation) in order to better understand this phenomenon.

Systemic inflammation is a hallmark of cancer [31] and has long been thought to play a pivotal role in cachexia’s pathophysiology. Numerous inflammatory biomarkers have been evaluated in the context of patients with cancer, and these have been consistently associated with poorer survival [32,33]. While contention remains regarding the driver of this inflammatory response, many believe it represents a host response to tumour hypoxia/necrosis, resulting in metabolic and neuroendocrine dysfunction and subsequent catabolism [34]. Alterations to haematopoiesis and the production of acute-phase proteins provide numerous potential blood markers of inflammation. The previous comparison of these, both in isolation and combination, has demonstrated that many of these have good prognostic utility; however, variation between markers is evident [35]. The NLR was selected as an appropriate marker of systemic inflammation for this cohort owing to its widespread use for prognostication [36,37] and ready availability. This is also a component of the ‘Cachexia Index’ (CXI) composite marker (skeletal muscle index, albumin, and NLR), which our group recently reported as being strongly associated with progression during neoadjuvant chemotherapy and poorer survival in patients with curatively treated oesophagogastric cancer [38]. When the constituent parts of CXI were considered in isolation in this previous study, the NLR was by far the most strongly associated with adverse outcomes. It can be argued that other inflammatory markers, such as C-Reactive Protein (CRP), would be preferable alternatives [39] given the widely validated prognostic effect of the Glasgow Prognostic Score (CRP and albumin) across numerous cancer populations; however, this is not routinely collected at the time of clinical staging in our practice. Future work to identify the optimum inflammatory biomarker for oesophagogastric patients would be valuable for the effective application of the GLIM criteria for cachexia.

The present study focussed on the classification of cachexia in Scottish patients with oesophagogastric cancer. Further investigation is required to establish the applicability of these findings in other tumour sites and different geographical locations. Variations in both host and disease characteristics have the potential to influence the prognostic ability of cachexia classifiers, and these vary markedly between cancer populations. Furthermore, cachectic phenotypes may vary, even between patients with the same cancer site [40], which presents additional difficulty for selecting one simple and pragmatic definition. Other limitations of this work include the retrospective use of routine clinical data and electronic patient records, which resulted in the exclusion of some patients from the sample who had missing datapoints. Within the present study, the majority of included patients were treated with palliative intent or best supportive care. As such, the subgroup of patients undergoing curative treatment was likely underpowered to compare cachexia classifiers. Future research focussed on those with earlier-stage disease may be fruitful, with the characterisation of cachexia being closer to its genesis. It could be argued that there is a greater scope for cachexia to influence clinical management in curatively treated disease, and targeted multimodal interventions are likely to be more feasible and effective in this setting. Beyond survival prognostication, the features of cachexia are known to influence patients’ quality of life and the efficacy of anti-cancer therapies, and patients should be counselled regarding this as part of the shared decision-making process. The presence of cachexia may also necessitate targeted multimodal treatments to lessen the effects of this devastating syndrome. This ‘host stage’ should be considered an adjunct to tumour stages that may inform the MDT and help guide their decision-making.

## 5. Conclusions

The results of the present study indicate that the identification of cancer cachexia using the GLIM criteria leads to improved prognostication in patients with oesophagogastric cancer compared to the Fearon et al. consensus definition. The consideration of systemic inflammation as an aetiological driver of wasting is key.

## Figures and Tables

**Figure 1 cancers-17-00448-f001:**
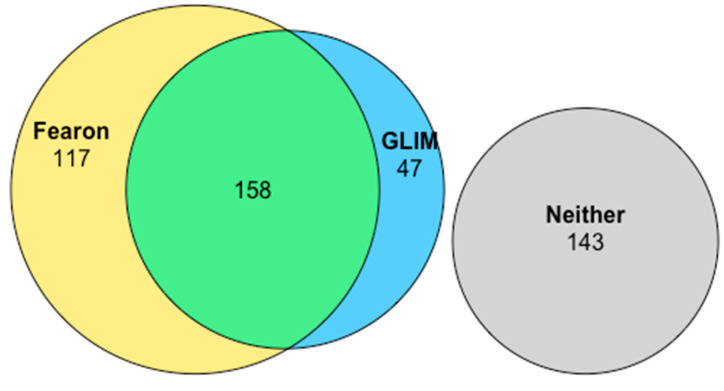
Euler diagram of overlap between cachexia definitions in oesophagogastric cancer.

**Figure 2 cancers-17-00448-f002:**
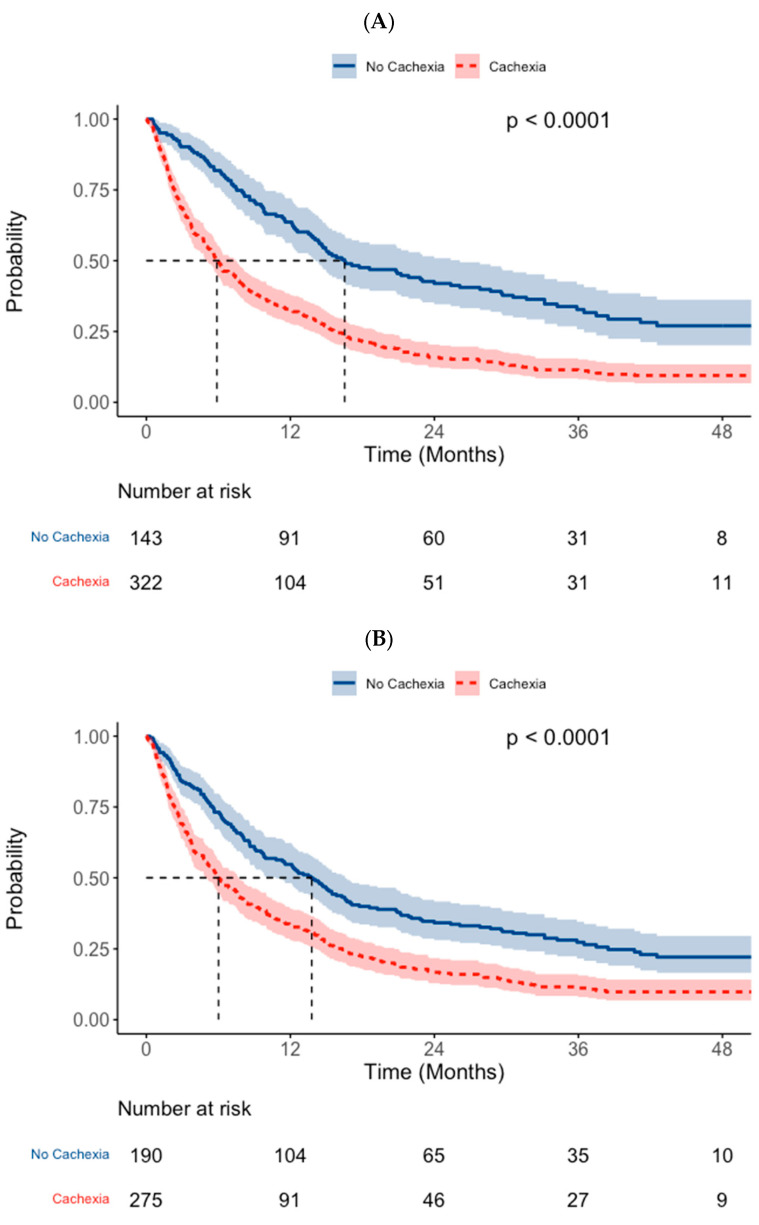
Comparison of overall survival between groups using cachexia classifiers: (**A**) cachexia (either definition) vs. non-cachectic, (**B**) Fearon et al.*-*defined cachexia vs. non-cachectic, (**C**) GLIM*-*defined cachexia vs. non-cachectic, and (**D**) combination of definitions.

**Table 1 cancers-17-00448-t001:** Clinicopathological characteristics at diagnosis compared by cachexia classifier.

		Fearon			GLIM		
		No Cachexia (*n* = 190)	Cachexia (*n* = 275)	*p* value	No Cachexia (*n* = 260)	Cachexia (*n* = 205)	*p* Value
Age	Median [IQR]	69.0 [62.0–78.0]	73.0 [64.0–80.0]	0.033	68 [60.0–78.0]	74.0 [67.0–81.0]	<0.001
Sex	Male	133 (70.0)	178 (64.7)	0.277	172 (66.2)	139 (67.8)	0.782
	Female	57 (30.0)	97 (35.5)		88 (33.8)	66 (32.2)	
ASA Grade	1	16 (8.4)	16 (5.8)	0.691	20 (7.7)	12 (5.9)	0.005
	2	83 (43.7)	121 (44.0)		131 (50.4)	73 (35.6)	
	3	85 (44.7)	131 (47.6)		103 (39.6)	113 (55.1)	
	4	6 (3.2)	7 (2.5)		6 (2.3)	7 (3.4)	
Charlson	0 to 1	122 (64.2)	177 (64.4)	0.681	182 (70.0)	117 (57.1)	0.012
Comorbidity	2 to 4	58 (30.5)	88 (32.0)		70 (26.9)	76 (37.1)	
Index	≥5	10 (5.3)	10 (3.6)		8 (3.1)	12 (5.9)	
ECOG-PS	0	80 (42.1)	89 (32.4)	0.112	118 (45.4)	51 (24.9)	<0.001
	1	64 (33.7)	96 (34.9)		93 (35.8)	67 (32.7)	
	2	27 (14.2)	62 (22.5)		33 (12.7)	56 (27.3)	
	3	18 (9.5)	25 (9.1)		15 (5.8)	28 (13.7)	
	4	1 (0.5)	3 (1.1)		1 (0.4)	3 (1.5)	
Height (m)	Median [IQR]	1.7 [1.6–1.8]	1.7 [1.6–1.8]	0.284	1.7 [1.6–1.8]	1.7 [1.6–1.8]	0.640
Weight (kg)	Median [IQR]	81.0 [70.0–95.0]	70.4 [58.0–84.8]	<0.001	78.0 [66.0–92.8]	71.0 [59.0–82.5]	<0.001
BMI (kg/m^2^)	Median [IQR]	28.0 [25.0–31.0]	25.0 [21.0–29.0]	<0.001	27.5 [24.0–31.0]	25.0 [21.0–28.0]	<0.001
Weight Loss (kg)	Median [IQR]	0.0 [0.0–0.0]	6.4 [4.2–11.8]	<0.001	2.0 [0.0–6.4]	5.0 [2.0–10.0]	<0.001
MUST Score	High	73 (38.4)	194 (70.5)	<0.001	138 (53.1)	129 (62.9)	0.024
	Medium	41 (21.6)	30 (10.9)		38 (14.6)	33 (16.1)	
	Low	76 (40.0)	51 (18.5)		84 (32.3)	43 (21.0)	
Tumour Site	Oesophagus	150 (78.9)	185 (67.3)	0.008	186 (71.5)	149 (72.7)	0.866
	Stomach	40 (21.6)	90 (32.7)		74 (28.5)	56 (27.3)	
Histology	AC	145 (76.3)	196 (71.3)	0.506	200 (76.9)	141 (68.8)	0.164
	SCC	35 (18.4)	57 (20.7)		45 (17.3)	47 (22.9)	
	Missing (%)	10 (5.3)	21 (8.0)		15 (5.8)	17 (8.3)	
Clinical	T1	0 (0.0)	2 (0.7)	0.022	2 (0.8)	0 (0.0)	<0.001
Tumour	T2	1 (0.5)	3 (1.1)		3 (1.2)	1 (0.5)	
Stage (cT)	T3	154 (81.1)	196 (71.3)		209 (80.4)	141 (68.8)	
	T4	20 (10.5)	58 (21.1)		26 (10.0)	52 (25.4)	
	TX	15 (7.9)	16 (5.8)		20 (7.7)	11 (5.4)	
Clinical Node	N0	28 (14.7)	46 (16.7)	0.164	51 (19.6)	23 (11.2)	0.001
Stage (cN)	N1	57 (30.0)	57 (20.7)		73 (28.1)	41 (20.0)	
	N2	50 (26.3)	74 (26.9)		70 (26.9)	54 (26.3)	
	N3	53 (27.9)	91 (33.1)		62 (23.8)	82 (40.0)	
	NX	2 (1.1)	7 (2.5)		4 (1.5)	5 (2.4)	
Clinical	M0	131 (68.9)	159 (57.8)	0.052	188 (72.3)	102 (49.8)	<0.001
Metastasis	M1	58 (30.5)	114 (41.5)		70 (26.9)	102 (49.8)	
Stage (cM)	MX	1 (0.5)	2 (0.7)		2 (0.8)	1 (0.5)	

Data are displayed as numbers (%) unless stated otherwise. ASA: American Society of Anaesthesiologists. ECOG-PS: Eastern Cooperative Oncology Group—Performance Status. BMI: Body Mass Index. MUST: Malnutrition Universal Screening Tool. AC: Adenocarcinoma. SCC: Squamous Cell Carcinoma.

**Table 2 cancers-17-00448-t002:** Combinations of cachexia classifiers and overall survival.

Combination of Classifiers		No. of Patients (%)	Median Survival (days)	95% Confidence Intervals
Fearon Cachexia	GLIM Cachexia			
No	No	143 (30.7)	503	428–795
Yes	No	117 (25.2)	363	248–485
Either		322 (69.2)	179	152–226
No	Yes	47 (10.1)	168	100–260
Yes	Yes	158 (34.0)	120	99–171

**Table 3 cancers-17-00448-t003:** Cox proportional hazard model for overall survival (Fearon et al.).

		All *n* (%)	Univariable HR (95% CI)	*p* Value	Multivariable aHR (95% CI)	*p* Value
Age	<50	30 (6.5)	-	-	-	-
(years)	50–60	64 (13.8)	0.70 (0.43–1.13)	0.148	0.91 (0.56–1.49)	0.719
	60–70	131 (28.2)	0.75 (0.49–1.16)	0.200	0.85 (0.54–1.33)	0.472
	70–80	141 (30.3)	1.06 (0.69–1.62)	0.789	1.03 (0.66–1.61)	0.884
	80–90	89 (19.1)	1.81 (1.17–2.82)	0.008	2.30 (1.41–3.77)	<0.001
	>90	10 (2.2)	1.54 (0.72–3.30)	0.262	1.88 (0.83–4.25)	0.131
Sex	Male	311 (66.9)	-	-	-	-
	Female	154 (33.1)	0.94 (0.76–1.16)	0.586	-	-
ASA Grade	1	32 (6.9)	-	-	-	-
	2	204 (43.9)	0.70 (0.47–1.04)	0.078	-	-
	3	216 (46.5)	1.17 (0.79–1.74)	0.432	-	-
	4	13 (2.8)	1.69 (0.88–3.27)	0.118	-	-
Charlson	0 to 1	299 (64.3)	-	-	-	-
Comorbidity	2 to 4	146 (31.4)	1.37 (1.11–1.70)	0.004	1.19 (0.94–1.50)	0.153
Index	≥5	20 (4.3)	1.98 (1.25–3.13)	0.004	1.26 (0.77–2.06)	0.364
ECOG-PS	0	169 (36.3)	-	-	-	-
	1	160 (34.4)	1.46 (1.14–1.86)	0.002	1.22 (0.95–1.58)	0.126
	2	89 (19.1)	2.74 (2.07–3.62)	<0.001	2.11 (1.54–2.88)	<0.001
	3	43 (9.2)	3.26 (2.27–4.67)	<0.001	2.48 (1.64–3.76)	<0.001
	4	4 (0.9)	11.95 (4.34–32.86)	<0.001	8.72 (3.03–25.06)	<0.001
Tumour	Oesophagus	335 (72.0)	-	-	-	-
Site	Stomach	130 (28.0)	1.08 (0.87–1.35)	0.491	0.86 (0.68–1.10)	0.236
Histology	AC	341 (78.8)	-	-	-	-
	SCC	92 (21.2)	1.21 (0.94–1.55)	0.138	-	-
Clinical Tumour	TX/T1–2	37 (8.0)	-	-	-	-
Stage (cT)	T3	350 (75.3)	2.71 (0.38–19.34)	0.319	1.37 (0.85–2.19)	0.195
	T4	78 (16.8)	5.64 (0.78–40.63)	0.086	2.12 (1.23–3.64)	0.007
Clinical Node	N0	3 (0.7)	-	-	-	-
Stage (cN)	N1	114 (24.5)	1.35 (0.96–1.90)	0.087	1.34 (0.93–1.93)	0.112
	N2	124 (26.7)	1.75 (1.25–2.45)	0.001	1.60 (1.10–2.35)	0.015
	N3	144 (31.0)	2.71 (1.96–3.75)	<0.001	1.93 (1.30–2.85)	0.001
	NX	9 (1.9)	5.96 (2.91–12.19)	<0.001	6.15 (2.58–14.69)	<0.001
Clinical	M0	290 (62.4)	-	-	-	-
Metastasis	M1	172 (37.0)	3.18 (2.57–3.92)	<0.001	2.71 (2.12–3.48)	<0.001
Stage (cM)	MX	3 (0.6)	9.25 (2.92–29.36)	<0.001	4.55 (1.04–19.91)	0.045
Cachexia	No	190 (40.9)	-	-	-	-
(Fearon et al.)	Yes	275 (59.1)	1.73 (1.40–2.13)	<0.001	1.41 (1.13–1.76)	0.002

HR: Hazard Ratio, displayed with 95% confidence interval. aHR: adjusted Hazard Ratio, following imputation, displayed with 95% confidence interval. ASA: American Society of Anaesthesiologists. ECOG-PS: Eastern Cooperative Oncology Group—Performance Status. Clinical Staging was conducted as per American Joint Committee on Cancer (AJCC) groupings.

**Table 4 cancers-17-00448-t004:** Cox proportional hazard model for overall survival (GLIM).

		All *n* (%)	Univariable HR (95% CI)	*p* Value	Multivariable aHR (95% CI)	*p* Value
Age	<50	30 (6.5)	-	-	-	-
(years)	50–60	64 (13.8)	0.70 (0.43–1.13)	0.148	0.87 (0.53–1.42)	0.576
	60–70	131 (28.2)	0.75 (0.49–1.16)	0.200	0.79 (0.50–1.23)	0.293
	70–80	141 (30.3)	1.06 (0.69–1.62)	0.789	0.92 (0.59–1.44)	0.717
	80–90	89 (19.1)	1.81 (1.17–2.82)	0.008	2.12 (1.30–3.49)	0.003
	>90	10 (2.2)	1.54 (0.72–3.30)	0.262	1.72 (0.77–3.86)	0.189
Sex	Male	311 (66.9)	-	-	-	-
	Female	154 (33.1)	0.94 (0.76–1.16)	0.586	-	-
ASA Grade	1	32 (6.9)	-	-	-	-
	2	204 (43.9)	0.70 (0.47–1.04)	0.078	-	-
	3	216 (46.5)	1.17 (0.79–1.74)	0.432	-	-
	4	13 (2.8)	1.69 (0.88–3.27)	0.118	-	-
Charlson	0 to 1	299 (64.3)	-	-	-	-
Comorbidity	2 to 4	146 (31.4)	1.37 (1.11–1.70)	0.004	1.14 (0.90–1.43)	0.282
Index	≥5	20 (4.3)	1.98 (1.25–3.13)	0.004	1.14 (0.70–1.86)	0.601
ECOG-PS	0	169 (36.3)	-	-	-	-
	1	160 (34.4)	1.46 (1.14–1.86)	0.002	1.30 (1.00–1.68)	0.049
	2	89 (19.1)	2.74 (2.07–3.62)	<0.001	2.12 (1.55–2.91)	<0.001
	3	43 (9.2)	3.26 (2.27–4.67)	<0.001	2.36 (1.56–3.56)	<0.001
	4	4 (0.9)	11.95 (4.34–32.86)	<0.001	8.40 (2.92–24.19)	<0.001
Tumour	Oesophagus	335 (72.0)	-	-	-	-
Site	Stomach	130 (28.0)	1.08 (0.87–1.35)	0.491	0.93 (0.73–1.19)	0.566
Histology	AC	341 (78.8)	-	-	-	-
	SCC	92 (21.2)	1.21 (0.94–1.55)	0.138	-	-
Clinical Tumour	TX/T1–2	37 (8.0)	-	-	-	-
Stage (cT)	T3	350 (75.3)	2.71 (0.38–19.34)	0.319	1.37 (0.86–2.21)	0.566
	T4	78 (16.8)	5.64 (0.78–40.63)	0.086	2.05 (1.19–3.52)	0.010
Clinical Node	N0	3 (0.7)	-	-	-	-
Stage (cN)	N1	114 (24.5)	1.35 (0.96–1.90)	0.087	1.35 (0.94–1.94)	0.102
	N2	124 (26.7)	1.75 (1.25–2.45)	0.001	1.57 (1.07–2.29)	0.021
	N3	144 (31.0)	2.71 (1.96–3.75)	<0.001	1.84 (1.24–2.73)	0.002
	NX	9 (1.9)	5.96 (2.91–12.19)	<0.001	5.53 (2.31–13.22)	<0.001
Clinical	M0	290 (62.4)	-	-	-	-
Metastasis	M1	172 (37.0)	3.18 (2.57–3.92)	<0.001	2.66 (2.08–3.41)	<0.001
Stage (cM)	MX	3 (0.6)	9.25 (2.92–29.36)	<0.001	5.25 (1.19–23.10)	0.028
Cachexia	No	260 (55.9)	-	-	-	-
(GLIM Criteria)	Yes	205 (44.1)	2.36 (1.93–2.89)	<0.001	1.57 (1.25–1.96)	<0.001

HR: Hazard Ratio, displayed with 95% confidence interval. aHR: adjusted Hazard Ratio, following imputation, displayed with 95% confidence interval. ASA: American Society of Anaesthesiologists. ECOG-PS: Eastern Cooperative Oncology Group—Performance Status. Clinical Staging was conducted as per American Joint Committee on Cancer (AJCC) groupings.

## Data Availability

Anonymised data may be available from the corresponding author upon reasonable request.

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
