# Peer review of "A Comparison of Established Diagnostic Criteria for Cachexia and Their Impacts on Prognostication in Patients with Oesophagogastric Cancer"

_cancers, 2025, doi:10.3390/cancers17030448_

Round 1

Reviewer 1 Report

Comments and Suggestions for Authors

The current manuscript highlights important differences in the diagnosis of cancer cachexia in patients affected by oesophagogastric cancers when using distinct diagnostic criteria, i.e. the Fearon (2011) and the GLIM (2019). The manuscript provides relevant new data, is timely and of high clinical value for oncologists, nurses and supportive oncologists. The manuscript reads well and is likely to attract the attention of the scientific community.

I only have minor suggestions that the Editor and the Authors may consider to implement for delivering a more reasoned message.

Distinct diagnostic tools highlight a morbid condition that we commonly define cachexia, however, focusing on distinct aspects or thresholds that do not agree on the inclusion or exclusion of specific patients. Is it the same medical condition, considering that the agreement among the two definitions include only 49% of the patients? In details, being inflammation the main factor differentiating the Fearon from the GLIM criteria, can the authors briefly discuss beyond the impact on survival (where the GLIM seems to better isolate the two groups), also considering patient’s quality of life, medical need, costs for the healthcare system, potential response to anti-cancer and anti-cachexia therapeutics, etc? This would potentially help guide future choice of which diagnostic criterion to use according to the aim of the implementation. 

Author Response

Comment 1: The current manuscript highlights important differences in the diagnosis of cancer cachexia in patients affected by oesophagogastric cancers when using distinct diagnostic criteria, i.e. the Fearon (2011) and the GLIM (2019). The manuscript provides relevant new data, is timely and of high clinical value for oncologists, nurses and supportive oncologists. The manuscript reads well and is likely to attract the attention of the scientific community.

Reply 1: Thank you very much to Reviewer 1 for their kind comments and interest in our work.

Comment 2: Distinct diagnostic tools highlight a morbid condition that we commonly define cachexia, however, focusing on distinct aspects or thresholds that do not agree on the inclusion or exclusion of specific patients. Is it the same medical condition, considering that the agreement among the two definitions include only 49% of the patients? In details, being inflammation the main factor differentiating the Fearon from the GLIM criteria, can the authors briefly discuss beyond the impact on survival (where the GLIM seems to better isolate the two groups), also considering patient’s quality of life, medical need, costs for the healthcare system, potential response to anti-cancer and anti-cachexia therapeutics, etc? This would potentially help guide future choice of which diagnostic criterion to use according to the aim of the implementation. 

Reply 2: We agree entirely with this point and our paper tries to highlight that these two definitions likely capture two distinct, but overlapping, patient groups within the spectrum of adverse host responses to cancer (cachexia being one subtype of disease related malnutrition). We have updated the discussion (lines 475-479) to try to better describe the implications of cachexia for patients, beyond survival prognostication as you have suggested.

Reviewer 2 Report

Comments and Suggestions for Authors

First of all, I congratuate authors for this tremendous work. 
The article is fit according to the journal's aim and scope and also interesting to readers.
I only have a few minor suggestions because the article is ready for further processing.
1. Try to improve the abstract.
2. The ethical approval number should be mentioned.
3. You can improve the conclusion.

Comments on the Quality of English Language

The quality of English is good; it only need minor improvement

Author Response

Comment 1: First of all, I congratuate authors for this tremendous work. The article is fit according to the journal's aim and scope and also interesting to readers.

Reply 1: Many thanks to Reviewer 2 for their kind words and interest in our research.

Comment 2: Try to improve the abstract.

Reply 2: We have made further edits to the abstract that we hope have improved it.

Comment 3: The ethical approval number should be mentioned.

Reply 3: We have explained to the editor that this work had been deemed not to require formal NHS REC ethical approval as per the Health Research Authority. We did, however, require institutional approval from the Caldicott Guardian as mentioned at the start of the methods. We have added an additional statement to the methods to clarify this point (line 105-106) regarding REC approval.

Comment 4. You can improve the conclusion.

Reply 4: In line with this comment and those of reviewer 1, we have expanded the end of the conclusion to discuss the wider implications of this work on the quality of life experienced by patients with cancer and their treatment pathways.

Reviewer 3 Report

Comments and Suggestions for Authors

The authors’ study focuses on a comparison of established diagnostic criteria for cachexia and their impact on prognostication in patients with oesophagogastric cancer. The results of the study indicate that identification of cancer cachexia using  the GLIM criteria leads to improved prognostication in patients with oesophagogastric cancer compared to the Fearon et al. consensus definition.

Global Leadership Initiative in Malnutrition (GLIM) consensus recommended the application of GLIM criteria to diagnose malnutrition in patients with cachexia.

Cancer cachexia is a challenging multifactorial clinical syndrome associated with impaired physical function, reduced quality of life, and worse survival.  Therefore, adequate nutritional assessment is significant for patients with cancer cachexia for the guidance of its nutritional intervention. GLIM-defined malnutrition independently predicts worse long-term survival both in oesophageal and  gastric cancer patients with cachexia. 10.1186/s12885-024-11880-z

Many studies provide clear evidence that the agreed -upon criteria for diagnosis of malnutrition are highly relevant and each of them alone can predict adverse clinical outcomes.

This study highlights the widely used criteria for malnutrition diagnosis and therefore can be used to support the development of global standards of care that will promote improved outcomes.

I consider the manuscript is significant to this field,  it can be accepted in present form.

Author Response

Comment 1: The authors’ study focuses on a comparison of established diagnostic criteria for cachexia and their impact on prognostication in patients with oesophagogastric cancer. The results of the study indicate that identification of cancer cachexia using  the GLIM criteria leads to improved prognostication in patients with oesophagogastric cancer compared to the Fearon et al. consensus definition.

Global Leadership Initiative in Malnutrition (GLIM) consensus recommended the application of GLIM criteria to diagnose malnutrition in patients with cachexia.

Cancer cachexia is a challenging multifactorial clinical syndrome associated with impaired physical function, reduced quality of life, and worse survival.  Therefore, adequate nutritional assessment is significant for patients with cancer cachexia for the guidance of its nutritional intervention. GLIM-defined malnutrition independently predicts worse long-term survival both in oesophageal and  gastric cancer patients with cachexia. 10.1186/s12885-024-11880-z

Many studies provide clear evidence that the agreed -upon criteria for diagnosis of malnutrition are highly relevant and each of them alone can predict adverse clinical outcomes.

This study highlights the widely used criteria for malnutrition diagnosis and therefore can be used to support the development of global standards of care that will promote improved outcomes.

I consider the manuscript is significant to this field,  it can be accepted in present form

Reply 1: Many thanks to Reviewer 3 for their kind comments and interest in our research.